The DNA methylation level against the background of the genome size and t-heterochromatin content in some species of the genus Secale L

Kalinka Anna 1 2
Achrem Magdalena biolmagda@poczta.onet.pl 1 2
Poter Paulina 1
1 Department of Cell Biology, Faculty of Biology, University of Szczecin , Szczecin , Poland
2 Molecular Biology and Biotechnology Center, Faculty of Biology, University of Szczecin , Szczecin , Poland
Lazo Gerard
Electronic publication date: 2017 Jan 24
Publication date: 2017
Volume: 5
Electronic Location ID: e2889
Received 2016 Sep 17; Accepted 2016 Dec 8
Copyright: ©2017 Kalinka et al.
Copyright year: 2017
Copyright holder: Kalinka et al.
License: This is an open access article distributed under the terms of the Creative Commons Attribution License, which permits unrestricted use, distribution, reproduction and adaptation in any medium and for any purpose provided that it is properly attributed. For attribution, the original author(s), title, publication source (PeerJ) and either DOI or URL of the article must be cited.
License URL: https://creativecommons.org/licenses/by/4.0/

Keywords: t-heterochromatin, Secale, ELISA, Genome size, Methylation DNA, MSAP

Funding: The authors received no funding for this work.

==============================
Methylation of cytosine in DNA is one of the most important epigenetic modifications in eukaryotes and plays a crucial role in the regulation of gene activity and the maintenance of genomic integrity. DNA methylation and other epigenetic mechanisms affect the development, differentiation or the response of plants to biotic and abiotic stress. This study compared the level of methylation of cytosines on a global (ELISA) and genomic scale (MSAP) between the species of the genus Secale. We analyzed whether the interspecific variation of cytosine methylation was associated with the size of the genome (C-value) and the content of telomeric heterochromatin. MSAP analysis showed that S. sylvestre was the most distinct species among the studied rye taxa; however, the results clearly indicated that these differences were not statistically significant. The total methylation level of the studied loci was very similar in all taxa and ranged from 60% in S. strictum ssp. africanum to 66% in S. cereale ssp. segetale, which confirmed the lack of significant differences in the sequence methylation pattern between the pairs of rye taxa. The level of global cytosine methylation in the DNA was not significantly associated with the content of t-heterochromatin and did not overlap with the existing taxonomic rye relationships. The highest content of 5-methylcytosine was found in S. cereale ssp. segetale (83%), while very low in S. strictum ssp. strictum (53%), which was significantly different from the methylation state of all taxa, except for S. sylvestre. The other studied taxa of rye had a similar level of methylated cytosine ranging from 66.42% (S. vavilovii) to 74.41% in (S. cereale ssp. afghanicum). The results obtained in this study are evidence that the percentage of methylated cytosine cannot be inferred solely based on the genome size or t-heterochromatin. This is a significantly more complex issue.

Introduction

Among epigenetic mechanisms, DNA methylation, histone modifications, chromatin or RNAi rearrangements can be distinguished, which are responsible for the regulation of gene expression and the maintenance of genomic integrity. All these mechanisms can affect the phenotype of the organism without altering the DNA sequence (Grant-Downton & Dickinson, 2005; Jablonka & Raz, 2009; Hirsch, Baumberger & Grossniklaus, 2013; Alonso et al., 2015). In plants, changes in the level of cytosine methylation may affect the viability, fertility, inbred status or herbicide resistance (Sano et al., 1990; Johannes et al., 2009; Verhoeven et al., 2010; Herrera & Bazaga, 2011; Vergeer, Wagemaker & Ouborg, 2012).

Epigenetic mechanisms, particularly through DNA methylation, may also affect macroevolution by the potential impact on diversification rate and speciation. In addition, methylated cytosines are more frequently mutated than the unmethylated ones (Nachman & Crowell, 2000; Ossowski et al., 2010; Alonso et al., 2015). Therefore, methylation can influence the frequency of point mutations and favor the occurrence of evolutionary changes (McClintock, 1984; Gorelick, 2003). The analysis of the level of global methylation of cytosines is the first step in evaluating the role of this epigenetic mechanism in the evolution of non-model organisms (Rozhon et al., 2008). Interspecific variation in the cytosine global methylation is not well understood, although some studies showed significant differences between species (Finnegan et al., 1998; Bender, 2004; Alonso et al., 2015). There is no information available whether interspecies differences in the level of cytosine global methylation are associated with changes resulting from their evolution, e.g., the size of the genome. The size of plant genome is a species-specific trait, and its enlargement in the course of evolution resulted primarily from an increase in the number of copies of repetitive sequences (Bennett & Leitch, 2012; Leitch & Leitch, 2013; Alonso et al., 2015) thus, theoretically, one could expect an increase in the level of methylation. Repetitive sequences are mainly present in the heterochromatin regions. It is within heterochromatin areas, such as telomeric, centromeric and pericentromeric regions (Li et al., 2008; Zilberman et al., 2007) where high levels of DNA methylation are observed, while they are lower in the euchromatin. Transposable elements (TEs), the amplification of which may lead to an increase in the size of the genome are such repeated sequences. Usually they are highly methylated, thus they can not be active, which protects genome integrity (Fujimoto et al., 2008; Sasaki et al., 2011; Kawanabe et al., 2012). Under stress conditions, silencing of TEs can be abrogated, they can transpose, which especially in the case of retroelements can cause a significant increase in the number of copies in the genome. After stress cessation, they are silenced again. If a TE is inserted in the coding region, it can inhibit the expression of a given gene (Kakutani, 2002; Tsukahara et al., 2009) or neighboring genes (Fernandez et al., 2010; Fujimoto et al., 2006; Hollister & Gaut, 2009; Lippman et al., 2004; Naito et al., 2009), because the heterochromatin state can spread to adjacent regions. Through this mechanism, genome expands and the level of global methylation changes (Bird, 1995; Federoff, 2000; Federoff, 2000; Grover & Wendel, 2010). Since the distribution of transposable elements varies even among closely related species, it can be assumed that TE polymorphisms affect the epigenetic variability (Saze & Kakutani, 2007; Kawanabe et al., 2012).

There is an assumption that in rye evolution, from the time when Secale diverged from the common ancestor with wheat, there was an increase in the DNA content, which was associated with the addition of heterochromatin. It resulted in the formation of multiple copies of the same DNA fragments (Bennett & Smith, 1976). Comparisons of repetitive DNA sequences suggested that rye, barley, wheat and oat had a common ancestor, and therefore, at the beginning of the 70 sof the twentieth century scientists began to determine the phylogenetic relationship and interspecies similarity in the genus Secale (Flavell, Rimpau & Smith, 1977). Despite years of research on the genus Secale, they have not been fully determined. The reason for this may be that it is believed that in rye many evolutionary pathways exist (Tarkowski & Miazga, 1983). Hence, there is a need to search for different methods to verify the proposed systems of classification and phylogenetic relationships. It is now believed that studying the pattern of DNA methylation is very important in explaining genome (Kalisz & Purugganan, 2004). As mentioned above, the evolution of the genus Secale proceeded with increasing amounts of heterochromatin, mainly telomeric heterochromatin. This is clearly correlated with the size of their genomes. Secale sylvestre has the lowest amount of t-heterochromatin (Singh & Röbbelen, 1977), and has the smallest genome (7.23 pg) (Bennett & Smith, 1976) among the species of rye. Secale cereale is considered evolutionarily the youngest species, having the most t-heterochromatin (Singh & Röbbelen, 1977) and the largest genome (8.28 pg) (Bennett & Smith, 1976) among Secale. It is then interesting whether the results of DNA methylation will confirm this relationship or, perhaps, they will demonstrate the complexity of the process of DNA methylation and indicate that the evolution of rye is not solely based on heterochromatin formation but is associated with a different methylation, which results in significant phenotypic differences between species.

The MSAP (Methylation-Sensitive Amplification Polymorphism) method was used to compare the methylation status of the DNA fragments on a genomic scale between rye taxa. This technique is based on the use of methylation-sensitive enzymes and the selective amplification of restriction fragments obtained. MSAP combines the advantages of AFLP, which allows to visualize multiple markers per sample, and to analyze cytosine methylation at the restriction site (Baurens et al., 2003). MSAP is one of the latest methods, which has already been used in studies of genome differentiation in cotton, banana, barley, Arabidopsis, tobacco and wild emmer wheat (Baurens et al., 2003; Li et al., 2009; Yang & Gaut, 2011). ELISA was another method used to determine the level of global methylation in rye.

We analyzed the data concerning global (ELISA) and genomic scale cytosine methylation (MSAP) in eight taxa of the genus Secale, together with information on the size of their genome and the amount of t-heterochromatin. The aim of the research was to answer the question on whether the cytosine methylation variation was consistent with phylogenetic relationships in the genus Secale and whether it was evolutionarily related to the size of the genome and the amount of telomeric heterochromatin.

Materials and Methods

Plant material

The following taxa of Secale were included in the study: Secale cereale L. cv. Dankowskie Zielonkawe, Secale cereale ssp. afghanicum, Secale cereale ssp. segetale, Secale strictum, Secale strictum ssp. africanum, Secale strictum ssp. kuprijanovii, Secale sylvestre and Secale vavilovii.

Caryopses of all taxa mentioned above were obtained from Botanical Garden of Polish Academy of Science (Warsaw, Poland).

DNA isolation

Total genomic DNA was isolated from 7-day-old seedlings using the DNeasy Plant Mini Kit (Qiagen, Valencia, USA) according to the manufacturer’s instruction. Each DNA extract was prepared from 30 randomly selected seedlings. The quality and quantity of the DNA samples was assessed by 0.8% agarose electrophoresis in 1×TBE buffer and measurements done using Nanodrop 2000c spectrophotometer (Thermo Scientific, Vilnius, Lithuania).

MSAP

500 ng of genomic DNA was digested with 1 µl of FastDigest® EcoRI (Fermentas, Vilnius, Lithuania) in a total volume of 50 µl for 20 min at 37°C. After EcoRI inactivation for 5 min at 80°C each sample was divided into two separate series: one for HpaII, one for. The DNA samples were digested with 1 µl of FastDigest® HpaII (Fermentas) and 1 µl of FastDigest® MspI (Fermentas) in a total volume of 50 µl for 30 min at 37°C. Enzymes were inactivated by incubation for 15 min at 65°C. EcoRI and HpaII/MspI adapters (Table 1) were prepared by mixing 50 µM oligonucleotides in equal volumes and incubation at the following profile: 10 min at 65°C, 10 min at 37°C, 10 min at 25°C, 10 min at 4°C. The restricted DNA was added the ligation mixture (2 µM Eco and Hpa/Msp adapter, 1×ligase buffer, 0.2 mM dATP, 5U T4 DNA ligase (Fermentas)) and incubated for 3 h at 37°C. The samples were diluted ten times, and 5 µl of the diluted DNA was added to the preamplification mixture. The total volume of 50 µl of preamplification mixture also contained: 10 µM Hpa + A/C primer (Table 1), 10 µM Eco + A primer (Table 1), 1×polymerase buffer, 0.2 mM dNTP, 2.5 mM MgCl2, 2 U DNA Polymerase (DyNAzyme™, Finnzymes, Espoo, Finland). The PCR conditions were as follows: initial denaturation of 1 min at 94°C; 25 cycles—30 s at 94°C, 60 s at 60°C, 60 s at 72°C; final elongation of 5 min at 72 °C. The PCR products were diluted 20 times and 5 µl of the diluted sample was added to 15 µl of selective amplification mixture. The final concentrations of the components in the mixture were: 4 µM Hpa selective primer (Table 1), 4 µM Eco selective primer (Table 1), 1 × polymerase buffer, 0.2 mM dNTP, 2.5 mM MgCl2, 0.5 U DNA Polymerase (DyNAzyme™, Finnzymes). The PCR conditions were as follows: initial denaturation of 1 min at 94°C; 12 cycles—30 s at 94°C, 30 s at 60°C, 60 s at 72°C; 22 cycles—30 s at 94°C, 30 s at 56 °C, 60 s at 72°C final elongation of 30 min at 72°C. In total 35 different combinations of selective primers were tested, out of which nine combinations were chosen for the final analysis (Table 1).

Table 1 The sequences of the oligonucleotides used in this study and the combinations of selective primers.

Type	Name/combination	Sequences	
Adapters	Eco adapter	5′-CTCGTAGACTGCGTACC and	
		5′-AATTGGTACGCAGTCTAC	
	Hpa/Msp adapter	5′-GATCATGAGTCCTGCT and	
		5′-CGAGCAGGACTCATGA	
Preselective primers	Eco+A	5′-GACTGCGTACCAATTCA	
	Hpa+A	5′-ATCATGAGTCCTGCTCGGA	
	Hpa+C	5′-ATCATGAGTCCTGCTCGGC	
Selective primers	Eco+AC	5′-GACTGCGTACCAATTCAC	
	Eco+AG	5′-GACTGCGTACCAATTCAG	
	Hpa+ATG	5′-ATCATGAGTCCTGCTCGGATG	
	Hpa+ATT	5′-ATCATGAGTCCTGCTCGGATT	
	Hpa+ACC	5′-ATCATGAGTCCTGCTCGGACC	
	Hpa+ACG	5′-ATCATGAGTCCTGCTCGGACG	
	Hpa+AAG	5′-ATCATGAGTCCTGCTCGGAAG	
	Hpa+CTT	5′-ATCATGAGTCCTGCTCGGCTT	
	Hpa+CGC	5′-ATCATGAGTCCTGCTCGGCGC	
Selective primers pairs	Eco+AC and Hpa+ATG; Eco+AG and Hpa+ATG; Eco+AC and Hpa+ATT; Eco+AC and Hpa+ACC; Eco+AC and Hpa+ACG; Eco+AC and Hpa+AAG; Eco+AG and Hpa+CTT; Eco+AC and Hpa+CGC; Eco+AC and Hpa+CTT	

PCR products were separated on a 6.5% denaturing polyacrylamide gel. 100 ml of the gel was prepared from RapidGel™-XL-6% Liquid Acrylamide (Affymetrix, Cleveland, USA) containing 6% modified acrylamides, 7 M Urea, 89 mM Tris, 89 mM boric acid and 2 mM EDTA, and to this solution 1,000 µl of 10% APS and 100 µl of TEMED was added. Electrophoresis in 1×TBE buffer was carried out on the Sequi-Gen GT Sequencing Cell (Bio-Rad, Hercules, USA) with the Power Pac™(Bio-Rad) as a power supply unit. The gel was pre-electrophoresed for 1 h (100 W, 2,000 V, 50 mA). Loading buffer (98% formamide, 20 mM EDTA, 0,01% bromophenol blue, 0,01% xylene cyanol) was added to the samples (12 µl per 20 µl), the mixture was denatured at 95 °C for 5 min, placed in ice for 10 min. 7 µl of the mixture from each samples were loaded in the wells. Simultaneously with the samples the pBR322 DNA/BsuRI (HaeIII) marker (Fermentas) was prepared and loaded onto gel. Electrophoresis was carried out for 2.5 h (100 W, 2,000 V, 50 mA). After electrophoresis bands were detected by silver staining technique using Silver Sequence™ DNA Sequencing System (Promega, Madison, USA), according to manufacturer’s protocol. The documentation and analysis of the gels were done by means of GS-800 Calibrated Densitometer (Bio-Rad) and Quantity One® ver. 4.6.9 (Bio-Rad) software.

MSAP procedure was repeated—three biological replicates were performed. Reproducible, clearly distinguishable fragments were scored in a form of binary matrix as either present (1) or absent (0) for the samples. MSAP markers were classified as different types: 11, 01, 10, or 00 accordingly to the presence (1) or absence (0) of the corresponding amplified fragment in the digestion reactions with HpaII and MspI, respectively.

Assessment of methylation level (%) of Methylation-Susceptible Loci (MSL), Principal Coordinate Analyses (PCoA) and Neighbor-Joining tree were done by means of R program (R version 3.2.2) with msap package.

UPGMA (unweighted pair group method using arithmetic averages algorithm was used for cluster analysis in order to show the relationships between taxa. Dice Index was computed in FreeTree (Pavlícek, Hrdá & Flegr, 1999) program and the dendrogram was visualized in TreeView (Page, 1996) program. Bootstrapping with 1,000 repetition was done in Free Tree program. The data from double matrix including: 11, 01, 10 and 00 types were transformed into single 1/0 matrix, in which 1 represented any type of methylated fragment (types: 01, 10 or 00) and 0 represented unmethylated fragment (type 11).

The analysis of differences between the taxa (U-values) was based on the suggestion of Wang et al. (2014): p=y1+y2n1+n2;

q=1−p;

δp1−p2=√pq1∕n1+1∕n2;

U=p1−p2δp1−p2

where n1 represents the total number of fragments of a given sample, n2 the total fragments of another sample, y1 the total number of methylated fragments of a given sample values, y2 the total number of methylated fragments in another sample, p1 the proportion (%) of methylated fragments of a given sample values and p2 the proportion (%) of methylated fragments in another sample.

Giemsa staining

Roots from 2-day-old seedlings were treated with 0.05% colchicine for 3 h at room temperature in the dark prior to fixation in ethanol-glacial acetic acid (3:1). Roots were macerated in a mixture of 4% (w/v) pectinase (Fluka), 6% (w/v) hemicellulase (Sigma-Aldrich, St. Louis, USA) and 4% (w/v) cellulase (Sigma-Aldrich) in 0.01 M citric acid—sodium citrate buffer (pH 4.8), for 3 h at 37°C . Roots were washed in distilled water. Each root tip were dissected on a single microscope slide, a drop of 45% acetic acid was added and the root tip was squashed under a cover glass. Preparations were heated for 20 min at 47 °C. The cover slip was removed after freezing over dry ice, and the slides were air-dried overnight. Chromosomes were stained with Giemsa reagent according to Darvey & Gustafson (1975) and Merker (1976). Preparations were analyzed under Eclipse E600 microscope (Nikon, Tokyo, Japan). NIS Elements ver. 3.00 SP7 (Nikon) software was used to measure: total chromosomes lengths, arms lengths and telomeric C-bands lengths. Based on the measurements the percentage of total telomeric heterochromatin content in the genome of each Secale taxon was computed. The Kruskal–Wallis test was performed in Statistica software ver. 12.

ELISA

The 5-mc DNA ELISA Kit (Zymo Research, Irvine, CA, USA) was used to detect global 5-methylocytosine in the DNA. 100 ng of each sample DNA was used in the assay. Four replicates of each sample were prepared in a single experiment, and the experiment was repeated three times. The analysis was performed according to the manufacturer’s instruction. Adequate controls included in the were used in the experiments. Plates were read in the Infinite® M200 Pro microplate reader (Tecan, Crailsheim, Germany). Magellan™ ver. 7.0 (Tecan) software was used to generate the standard curve and to quantify the percentage of 5-mC in DNA samples. The Tuckey test was performed in Statistica software ver. 12.

Flow cytometry—20 mg of coleoptiles from 7-day-old seedlings were chopped in 1 ml of ice-cold nuclei isolation buffer (45 mM MgCl2⋅6H2O, 20 mM MOPS, 0,1% (v/v) Triton X-100, 1% PVP; pH 7.0) supplemented with propidium iodide (Sigma-Aldrich; 50 µg/mL) and ribonuclease A (Sigma-Aldrich; 50 µg/mL). The homogenate was filtered through a 42-mm nylon mesh into a 1.5 ml tube and incubated for 10 min at ice. Secale cereale was used as an internal standard. The measurements were done at the rate 20–50 nuclei. For each sample, fluorescence in at least 7,000 was measured using BD FACSCalibur™ flow cytometer (Becton Dickinson, San Jose, CA, USA). The mass of the genome was calculated from the formula below: Sample2Cvaluepg=Reference2Cvalue×Sample2CmeanpeakpositionReference2Cmeanpeakposition.

Reference 2C value for Secale cereale = 16.55 pg (Bennett & Smith, 1976).

Histograms were analyzed using the CellQuest™ Pro (Becton Dickinson) software. The results were estimated using Tuckey test (Statistica software ver. 12).

Results

MSAP

Methylation profiling by MSAP relies on the use of methylation-sensitive enzymes. In this work we used a pair of enzymes, MspI and HpaII, which recognize the 5′-CCGG-3′ sequence, but they cut the DNA depending on the methylation state of that sequence: MspI will not cut the DNA if the external cytosine is fully methylated or hemimethylated. HpaII will not cut the DNA when the cytosine is fully methylated, and it cuts the DNA when the external cytosine is hemimethylated (Table 2). Therefore, the application of this pair of enzymes in MSAP allows to specify 4 different methylation states of the 5′-CCGG-3′ sequence: type 11—no methylation, type 01—methylated CpG sequence, type 10—hemimethylated sequence or methylated CpCpG sequence, type 00—hypermethylation (Table 2). In type 4, it is also possible that no band in a given taxon does not result from changes in the methylation state, the reason may be the lack of this sequence. Hence, the MSAP analysis may introduce some error. However, based on the conducted experiments, including less clear bands omitted in this analysis, it could be said that band patterns of the examined rye taxa were very similar to each other, and even if the analysis was burden with the above-described error, it was small. Before we performed the MSAP analysis, some preliminary tests with the classical AFLP technique had been done (A Kalinka, 2014, unpublished data). We calculated the Dice coefficient for each pair of species/subspecies. The values were high (0.89–0.95), thus we decided that MSAP analysis would be reliable. The values of the coefficient were not surprising as from ISSR and IRAP analyses, which allow to study the polymorphism of highly variable inter-microsatellite and inter-retrotransposon sequences, the values were not low, 0.48–0.85 (IRAP) and 0.39–0.82 (ISSR) (Achrem, Kalinka & Rogalska, 2014). We were also aware that MSAP can analyze a small quantity of methylated cytosines in the genome that is why we also complemented with global methylation data, which shed more light on the variation of methylation in Secale.

Table 2 The methylation status of the 5′-CCGG-3′ sequence base on the presence (1) or absence (0) of DNA restriction fragments (HpaII and MspI).

						HpaII	MspI	
C	C	G	G	type 11	Unmethylated	1	1	
G	G	C	C	
C	mC	G	G	type 01	Internal cytosine methylation	0	1	
G	G	mC	C	
mC	mC	G	G	type 10	Hemimethylated	1	0	
G	G	C	C	
or				
mC	C	G	G	
G	G	C	C	
mC	mC	G	G	type 00	Full methylation	0	0	
G	G	mC	mC	
Notes.

C Unmethylated cytosine

mC Methylated cytosine

To ensure the robustness of the results, the analysis included only the most evident and repetitive fragments in the size range 50–500 nt. In total, 149 loci were selected of all the reactions performed with 9 primer combinations. Among these loci, 98 methylation-susceptible loci (MSL) and 51 non-methylated loci (NML) were distinguished. The number of polymorphic MSL was 14 (14% of total MSL) and the NML was 7 (14% of total NML). The level of methylation of MSL loci is shown in Fig. 1, which demonstrates that type 01 (internal methylated cytosine) and type 00 (hypermethylation) fragments are most frequent among rye taxa, reaching approximately 43% and 38%, respectively. Type 11 fragments were definitely the least frequent (approximately 7%), and were followed by slightly more frequent type 10—(approx. 12%).

Figure 1 Methylation level (%) of Methylation-Susceptible Loci (MSL) in Secale taxons.

Type 1–unmethylated, type 2–internal cytosine methylation, type 3–hemimethylated, type 4–full methylation.

Both the MSL analysis alone (Fig. 2) and the combined analysis of MSL and NML loci (Fig. 3) showed that S. sylvestre was the most distinct from other species. Secale vavilovii also showed greater epigenetic variation in relation to the taxa belonging to the species S. cereale and S. strictum. The total methylation level of the studied loci was very similar in all taxa (Fig. 4). The U values were calculated between all taxa to verify whether the differences in the level of methylation are statistically significant. Higher U values indicated greater differences between taxa, but only U values >1.96 (U(0.05) = 1.96) would mean statistically significant differences. The U values in the studied rye taxa ranged between 0 for S. sylvestre and S. vavilovii, S. sylvestre and S. cereale, S. cereale and S. vavilovii and 1.25 for Secale s. ssp. africanum and S.c. ssp. segetale, indicating that the methylation pattern did not differ significantly between any pair of taxa (Table 3). Dice coefficients confirmed this high similarity (Table 3), as it ranged between pairs of individual taxa from 87% (S. sylvestre and S. s. ssp. kuprijanovii) to 96% (S. cereale and S. s. ssp. kuprijanovii, S. cereale and S.c. ssp. segetale, S.c. ssp. segetale and S. s. ssp. kuprijanovii). Since the rye taxa investigated in this study belonged to 4 species, it was very interesting how epigenetically similar to each other were subspecies within the species. Such analysis was conducted and the graphical representation is shown in Fig. 2. Figure 2A shows the variation between the taxa without grouping them in species, while Fig. 2B shows the variation between species. Dendrogram (Fig. 2C) shows the relationship between eight rye taxa grouped into four species. The results clearly indicated that epigenetic similarity between species was sufficiently significant so that the subspecies of S. cereale and S. strictum did not group together. The bootstrapping analysis confirmed this result (Fig. 3). Low values obtained indicated that the presented clustering of taxa was unreliable, and only S. sylvestre was definitely different from other taxa, but these differences were not statistically significant.

Figure 2 Principal Coordinate Analyses for epigenetic differentiation between the groups and Neighbor-Joining tree for epigenetic distances.

Principal Coordinate Analyses (PCoA) for epigenetic (MSL) differentiation between Secale taxons (A) and Secale species (B); (C): Neighbor-Joining tree of Secale species for epigenetic (MSL) distances; colors represent different taxons/species; (A), (B): C1 and C2 coordinates are shown with the percentage of variance explained by them, in (B) different point types represent subspecies from different species, species labels show the centroid for the points cloud in each species, ellipses represent the average dispersion of those points around their center, the long axis of the ellipse shows the direction of maximum dispersion and the short axis, the direction of minimum dispersion; cereale: S. cereale, S. c. ssp. afghanicum, S. c. ssp. segetale; strictum: S. strictum, S. s.ssp. africanum, S. s. ssp. kuprijanovii; sylvestre: S. sylvestre; vavilovii: S. vavilovii.

Figure 3 UPGMA dendrogram representing epigenetic relationships among rye taxons based on the data from MSAP analysis.

The numbers represent bootstrapping values.

Figure 4 The comparison of t-heterochromatin content and cytosine methylation.

Total telomeric heterochromatin content (%) in rye genomes, level (%) of cytosine methylation at CCGG sites from MSAP analysis, and global cytosine methylation (%) from ELISA analysis.

Table 3 The upper part of the table (grey cells) contains Dice coefficients (reflecting the epigenetic similarity among the rye taxons); the lower part (blue cells) contains the U-values (showing the differences between the rye taxons); both sets of data were calculated on the basis of MSL and NML loci polymorphism.

	S. c. ssp. afganicum	S. s. ssp. africanum	S. cereale	S. s. ssp. kuprijanovii	S.c. ssp. segetale	S. strictum	S. sylvestre	S. vavilovii	
S. c. ssp. afganicum		0.95	0.95	0.96	0.94	0.95	0.90	0.93	
S. s. ssp. africanum	0.35		0.92	0.92	0.90	0.93	0.88	0.93	
S. cereale	0.54	0.89		0.96	0.96	0.94	0.91	0.95	
S. s. ssp. kuprijanovii	0.36	0.71	0.18		0.96	0.95	0.87	0.92	
S. c. ssp. segetale	0.90	1.25	0.36	0.54		0.95	0.88	0.91	
S. strictum	0.18	0.53	0.36	0.18	0.72		0.88	0.89	
S. sylvestre	0.54	0.89	0.00	0.18	0.36	0.36		0.92	
S. vavilovii	0.54	0.89	0.00	0.18	0.36	0.36	0.00		

Giemsa staining

Thirty preparations of root tips of each rye taxa were prepared in order to determine the total percentage of telomeric heterochromatin. Five preparations were analyzed in detail, in which nine best metaphase plates were selected and the following measurements were performed in them: the overall length of each chromosome, short and long arm length, the length t-heterochromatin bands in individual chromosomes. On this basis, the total percentage of telomeric heterochromatin was calculated in the whole genome of each taxon. The results showed that the content of t-heterochromatin was varied (Fig. 4): the lowest amount of t-heterochromatin was found in S. sylvestre (6.18%) (Fig. 5B), S. s. ssp. africanum (6.25%), S. strictum (7.18%) and S.s. ssp. kuprijanovii (7.72%). No significant differences in the content of t-heterochromatin were found between these taxa (Table 4). More t-heterochromatin was present in the genomes of S. c. ssp. segetale (10.94%), S. c. ssp. afghanicum (11.90%), S. cereale (12.29%) and S. vavilovii (13.39%) (Fig. 5A). Also, in this case no significant differences were recorded between these taxa. Therefore, these Secale taxa could be divided into two groups, with a relatively lower and higher content of t-heterochromatin. In contrast to the MSAP method, the results of which showed that there are no significant differences between all taxa, and the taxa of the species S. cereale and S. strictum did not cluster together, the analysis of the content of t-heterochromatin confirmed the currently accepted taxonomic relationships. Admittedly, species that were the most different from others in the MSAP analysis, here also had the lowest (S. sylvestre; 6.18%) and the highest (S. vavilovii; 13.39%) t-heterochromatin contents. While the remaining taxa in the group with smaller amounts of heterochromatin-t belonged to the species S. strictum, and in the group with a higher t-heterochromatin content they belonged to the species S. cereale. Nevertheless, there were no statistically significant differences found in the t-heterochromatin content between S. s. ssp. kuprijanovii and S. c. ssp. segetale and S. c. ssp. afghanicum.

Figure 5 C-banded methaphase chromosomes of Secale, (A) Secale vavilovii, (B) Secale sylvestre.

Table 4 The upper part of the table (grey cells) contains P values from Kruskal–Wallis test for the differences in total t-heterochromatin content among the genomes of rye taxons; the lower part (blue cells) contains P values from Tukey test for the differences in total cytosine methylation among the genomes of rye taxons.

S. c. ssp. afganicum	S. s. ssp. africanum	S. cereale	S. s. ssp. kuprijanovii	S. c. ssp. segetale	S. strictum	S. sylvestre	S. vavilovii	
S. c. ssp. afganicum		0.000032*	1.000000	0.331233	1.000000	0.007999*	0.000004*	0.132453	
S. s. ssp. africanum	0.934268		0.000000*	0.526546	0.000016*	1.000000	1.000000	0.000000*	
S. cereale	0.951740	1.000000		0.002697*	1.000000	0.000015*	0.000000*	1.000000	
S. s. ssp. kuprijanovii	0.633481	0.998885	0.997694		0.226022	1.000000	0.175761	0.000003*	
S. c. ssp. segetale	0.456476	0.035269*	0.042688*	0.005712*		0.004762*	0.000002*	0.198283	
S. strictum	0.000137*	0.002251*	0.001789*	0.015542*	0.000118*		1.000000	0.000000*	
S. sylvestre	0.035658*	0.459018	0.414978	0.832923	0.000138*	0.444195		0,000000*	
S. vavilovii	0.530631	0.995110	0.991593	1.000000	0.003448*	0.024517*	0.899162		
Notes.

* Statistically significant differences (P < 0.05).

ELISA

In total, 12 measurements of the global cytosine content were performed in the genome of each of the rye taxa. Five taxa had a similar level of methylated cytosine (Fig. 4) and no significant differences were observed (Table 4) between S.c. ssp. afghanicum (74.41%), S. s. ssp. africanum (69.52%), S. cereale (69.80%), S. s. ssp. kuprijanovii (67.05%) and S. vavilovii (66.42%). Slightly lower level of methylation was detected in S. sylvestre (61.09%) and it only differed from the methylation state of S. c. ssp. afghanicum of the above-listed taxa. Very high content of 5-methylcytosine (82.86%) was found in S. c. ssp. segetale. In this case, it has a significantly higher level of methylation compared to the genomes of all the other taxa, with the exception of S. c. ssp. afghanicum. On the other hand, extremely low levels of DNA methylation was present in S. strictum (52.56%) and differed significantly from the methylation status of all taxa besides S. sylvestre.

The level of global cytosine methylation in DNA, as opposed to the total t-heterochromatin in the genome, did not overlap with the current taxonomic relationships. Within three taxa belonging to the species S. cereale: S. cereale, S. c. ssp. afghanicum and S. cereale differed significantly from S. c. ssp. segetale, while in the taxa belongong to S. strictum: subspecies S. strictum subsp. strictum were different from the other two subspecies S. s. ssp. africanum and S. s. ssp. kuprijanovii.

It is not surprising that the results of methylation analysis of the CCGG sequence by MSAP did not overlap with global analysis of cytosine methylation by ELISA, as in plant genomes, in addition to the methylation of CG sequences, CNG and CNN sequence methylation also occurs (Johnson et al., 2007; Meyer, 2011; González, Ricardi & Iusem, 2013). Thus, it was possible that the CpG methylation was similar in the genomes of all Secale taxa tested, while such high differences in the level of 5-methylcytosine were due to differences in the CNG and CNN methylation. In addition, the MSAP method allowed to examine only selected loci in a limited number, which might not give a complete picture of epigenetic variation.

Flow cytometry

The DNA content in the nucleus of the studied Secale taxa was the lowest in S. s. ssp. africanum (14.76 pg) and S. s. ssp. kuprijanovii (15.23 pg), in which it did not exceed 16 pg. All other taxa had the 2C value above 16 pg, with the largest amount of DNA measured in S. vavilovii (Fig. 6). It was this rye species that differed significantly from S. s. ssp. africanum (P < 0.001) and S. s. ssp. kuprijanovii (P = 0.007). In other cases, there were no significant differences in the DNA content in the nucleus (P > 0.05).

Figure 6 2C DNA nuclear content (pg).

* The 2C value for Secale cereale was used as reference value.

Discussion

DNA methylation is involved in many important processes in plants, including the regulation of gene expression, gene imprinting or growth and differentiation of plants (Finnegan et al., 1998). This is a dynamic process, differences in which depend on the degree of development or tissue type. The MSAP technique was used in the study to determine the polymorphism methylation level between species of the genus Secale. Methylation of the CCGG sequence was evaluated by using the combination of MspI and HpaII restriction enzymes. The level of methylation in the test species was high and ranged from 60–66%. Very similar methylation level was determined by MSAP in Triticum turgidum ssp. dicoccoides (Venetsky et al., 2015). However, using the same method, Shaked et al. (2001) found that the level of cytosine methylation in T. turgidum ssp. durum reached only 35%. The results showed a higher degree of DNA methylation in Secale taxa compared with the species of the genus Cycas (Sae-Eung et al., 2012) or species such as Brassica oleracea (Salmon et al., 2008), Elaeis guineensis (Jaligot et al., 2004) or Arabidopsis thaliana (Cervera, Ruiz-García & Martínez-Zapater, 2002), which may be caused by a higher number of repeated sequences (up to 92%) in the genomes of species of the genus Secale (Flavell, Rimpau & Smith, 1977).

Both the analysis of MSL alone and combined analysis of MSL and NML loci showed that S. sylvestre was the most epigenetically distinct species compared to other species studied, which was consistent with previous studies on the phylogeny of rye (Reddy, Appels & Baum, 1990; Del Pozo et al., 1995; De Bustos & Jouve, 2002; Chikmawati, Skovmand & Gustafson, 2005; Shang et al., 2006; Achrem, Kalinka & Rogalska, 2014). The dissimilarity of this species in relation to other Secale taxa was confirmed by molecular analysis of plastid DNA (Murai, Naiyu & Tsunewaki, 1989; Petersen & Doebley, 1993) repetitive sequences (Cuadrado & Jouve, 2002; Shang et al., 2006; Zhou et al., 2010; Achrem, Kalinka & Rogalska, 2014), 18S–5.8S–26S rDNA (De Bustos & Jouve, 2002) or AFLP (Chikmawati, Skovmand & Gustafson, 2005). The cited studies confirm the hypothesis of Singh & Röbbelen (1977) and Bennett, Gustafson & Smith (1977), that S. sylvestre is most likely phylogenetically the oldest among Secale species. Cuadrado & Jouve (2002), when examining the distribution of SSR sequences in the chromosomes of different taxa of rye, found that S. sylvestre separated earlier from the other species of the genus Secale. The lack of 480 bp family sequences in this species, indicated that this sequence could be amplified in the evolution of the genus Secale after the divergence of this taxon from a common ancestor. It can be concluded that the amplifications and deletions of repetitive sequences in the genomes of rye formed a basis of the evolutionary pathway of this genus (Cuadrado & Jouve, 2002), which also affected the level of DNA methylation. However, the differences between S. sylvestre and other rye taxa, demonstrated by the MSAP technique, were not statistically significant. It seems that genus Secale is not an exception. Epigenetic distance calculated on the basis of MSAP among three diploid Limonium species (L. nydeggeri, L. ovalifolium, L. lanceolatum) was very low. It ranged from 0.002 to 0.029 between pairs of individual species (Róis et al., 2013).

The higher epigenetic diversity of S. vavilovii compared to other rye taxa was surprising, however, as in the case of S. sylvestre, it was not statistically significant. The results confirmed that this species was very similar to S. cereale (Table 3), and confirmed the assumption that they might share a common ancestor. This thesis was also consistent with the study of Ren et al. (2011) and Bolibok-Bragoszewska et al. (2015), where this species was even classified as a subspecies of S. cereale. Osabe et al. (2014) observed different methylation levels at a low genetic variation in the genotypes of cotton, which were grown under the same environmental conditions over many generations. Similar results in other crops suggest the involvement of methylation changes, compensating for the absence of genetic variation (Fang et al., 2008; Fang et al., 2010).

The use of the MSAP method alone does not provide a complete methylation picture. This technique is limited by recognizing methylation only at the CCGG sites and not entirely reliable detection of hypermetylated regions. In plants, DNA methylation is not restricted to CpG islands only, as it occurs in three sequence contexts—CG, CHG and CHH (H = A, C or T) (Bender, 2004; Salmon et al., 2008; Kim et al., 2015), which are characterized by a specific pattern of inheritance and have their own regulatory pathways. However, numerous literature data suggest that the highest level of methylation is usually obtained in the CG context, intermediate in the CHG context, and the lowest level of methylation is observed in CHH (Feng et al., 2010; Eichten et al., 2016; Osabe et al., 2014; Salmon et al., 2008). On this basis, MSAP results can be considered to be meaningful when comparing epigenomes.

Another issue that should be raised here is that MSAP patterns might be in certain circumstances ambiguous Fulneček & Kovařík (2014). According to Fulneček, Matyasek & Kovařík (2002) or Cokus et al. (2008) the most frequent status of CCGG methylation is CmCGG, less frequent mCmCGG while mCCGG is the rarest. It is consistent with our results. However, as a consequence, in the regions where CCGG motifs are closely located, in of some HpaII-EcoRI fragments internal CmCGG might be located. The digestion of HpaII-EcoRI by MspI would cause to the misinterpretation of the MSAP pattern Fulneček & Kovařík (2014). Thus, as the interpretation of type 11 and type 01 patterns is not complicated, the patterns of 10 fragments may be more ambiguous. Considering this, we might expect, that the calculated methylation level based on MSAP, may not perfectly correspond to the actual methylation level of analyzed fragments. On the other hand, it did not affected the assessment of the Secale taxa similarity because the MSAP patterns were strikingly similar.

The ELISA technique was applied to complement MSAP results and elucidate global methylation in the species of the genus Secale. Determination of global methylation level of all cytosines in the DNA has its own drawbacks, as it does not provide information about the genomic position of methylated cytosines. However, knowledge of the global methylation may reflect functional changes in the genome, such as mutations, genomic stability, gene expression changes or chromosomal rearrangements (McClintock, 1984; Steward et al., 2002; Feschotte & Pritham, 2007; Rozhon et al., 2008; Bonchev & Parisod, 2013; Alonso et al., 2015). The level of global methylation turned out to be very high and ranged from 53% in S. strictum to 83% in S. c. ssp. segetale. In most species, as expected, it was higher than that obtained using MSAP, with the exception of species, such as S. strictum and S. sylvestre. It is possible that the CpG methylation was similar in the genomes of all Secale taxa tested, while such high differences in the level of 5-methylcytosine were due to differences in the CNG and CNN methylation. In addition, the MSAP method allows one to examine only selected loci in a limited number, which may not give a complete picture of epigenetic variation. The obtained variability in DNA methylation patterns among the tested taxa of rye was at a rather low level. However, it showed that in the process of evolution, each taxon accumulated epialleles and faithfully passed them on from generation to generation, which resulted in differences in the obtained percentages and changes in DNA methylation patterns between the taxa.

DNA methylation is an epigenetic mechanism that is directly related to the heterochromatin fraction in the genome. Thus, the evolutionary increase of genome sizes by heterochromatin addition affects the complexity of the genome and must be associated with increasing levels of global cytosine methylation in DNA (Fedoroff, 2012). This was confirmed by the study of Alonso et al. (2015) concerning the correlation between global methylation and the size of the genomes of 54 angiosperm species. The evolution of the species Secale proceeded with the addition of heterochromatin, especially telomeric heterochromatin, and consequently an increase in genome size. Therefore, in this study, the results of DNA methylation were complemented with the data on genome size and the amount of t-heterochromatin of individual taxa. This hypotheses were confirmed by the results of MSAP, and ELISA analyses, obtained in the taxon S. c. ssp. segetale. It was characterized by the highest methylation percentage, which positively correlated with a high DNA content—16.07 pg (2C), and large blocks of telomeric heterochromatin, which constituted as much as 11%. This relationship was similar in S. c. ssp. afghanicum and S. cereale. Hence, it would seem that with increasing genome size, the level of DNA methylation will also increase. However, the results obtained in this study are evidence that the percentage of methylated cytosine can not be inferred solely based on the genome size or t-heterochromatin. This is a significantly more complex issue. Alonso et al. (2015) demonstrated that methylation level increased slower than genome size. These authors concluded that during evolution when an increase in DNA content occurred, the content of methylated cytosine steadily decreased (Alonso et al., 2015). This may be due to the reduced percentage of repetitive sequences in the genome as compared with its size or due to methylation density of repetitive sequences. This study demonstrated that S. s. ssp. africanum had the smallest genome (2C–14.76 pg), and it also had the lowest percentage of telomeric heterochromatin (6%). However, in contrast to the expected value, the global methylation was high and reached 70%. This can be explained by the fact that it is an endemic species, which occurs only in South Africa and its evolution could be different compared to other rye taxa or subspecies S. strictum (Hammer, 1990). However, this example confirms that the increasing genome size in Secale does not necessarily means higher level of DNA methylation. An amount lower than the expected value of the global methylation was detected in S. vavilovii (66%) at high DNA (2C–16.66 pg) and t-heterochromatin contents (13%). This variation in the level of DNA methylation may be due to changes resulting from the adaptive process of plants, wherein the methylation is an important epigenetic mechanism (Sae-Eung et al., 2012). The process of DNA methylation/demethylation may affect the inhibition or activation of a gene or the entire range of different genes in different environment in order to adapt the species to changing biotic and abiotic conditions. Many literature data showed that changes in the level of genome methylation occurred even after short-term exposure of plants to pathogens, low temperature or drought (Steward et al., 2002; Lukens & Zhan, 2007; Peng & Zhang, 2009; Verhoeven et al., 2010; Grativol, Hemerly & Gomes Ferreira, 2012; Alonso et al., 2014). Therefore, differences in DNA methylation may be caused by the adaptation of organisms to a particular environment (Finnegan & Kovac, 2000). This was the conclusion drawn from MSAP and TMD (transposon methylation display) analyses of natural population of emmer wheat (T. turgidum ssp. dicoccoides), in which the level of DNA methylation was modified by the influence of the environmental factors (Venetsky et al., 2015). This also may be the reason of such a low level of global DNA methylation in S. strictum (53%), as compared to other taxa, and even its subspecies, despite the high DNA content (2C - 16.22 pg). The lack of relationships between the t-heterochromatin content and the level of global cytosine methylation in Secale may be also explained by the fact, that not only DNA methylation but also other epigenetic mechanisms are involved in maintaining the constitutive heterochromatin state (Soppe et al., 2002). The evidence comes from the studies on barley, in which differences in chromosomal distribution of constitutive heterochromatin and 5-mC pattern were found (Castiglione et al., 2008).

Additionally, genome size enlargement could occur in various species through the accumulation of other types of sequences, which would be associated with their different methylation levels. A large part of the genome size variation is caused by the differences in the presence and amplification of transposable elements, especially retrotransposons (Bennetzen et al., 1998; Pearce et al., 1996; Vicient et al., 2001; Schulman, Flavell & Ellis, 2004; Grover & Wendel, 2010). Mobile genetic elements are often the main target of DNA methylation in plants (Hirochika, Okamoto & Kakutani, 2000; Kato et al., 2003; Salmon et al., 2008), because they should be silenced to ensure the stability and integrity of the genome. Thus, changes in DNA methylation can be related to the presence or absence of transposable elements in the genome. It is associated with a different number of transposable elements, as well as the specificity of certain transposable elements that could be subject to amplification in the genome of a particular species after the evolutionary separation. It should be noted that epigenomic changes may occur under stress, particularly changes in the level of DNA methylation and activation of transposable elements. Especially in the case of retrotransposons it can be connected with a significant increase in genome size (Kim & Zilberman, 2014; Mirouze & Vitte, 2014; Hollister et al., 2011; Eichten et al., 2012; Eichten et al., 2013; Eichten et al., 2016). Although many examples show variation resulting from the absence or presence of TEs, leading to changes in DNA methylation (Hollister & Gaut, 2009; Hollister et al., 2011; Eichten et al., 2012; Eichten et al., 2013; Eichten et al., 2016; Ahmed et al., 2011), there are studies showing a minimal relationship between the variation of transposable elements and methylation (Li et al., 2015). They may result for example from the fact that insertions and deletions of TEs concern heterochromatin regions, leading to minimal changes in overall DNA methylation patterns (Eichten et al., 2016).

It is believed that since the genetic activity of TEs in plant is strictly dependent on their methylation, reducing their methylation with expanding genome size may increase the chances of genetic, phenotypic or evolutionary effects of TE action in plants with larger genomes (Banks & Fedoroff, 1989; Bonchev & Parisod, 2013; Alonso et al., 2015).

Conclusions

The genome enlargment and heterochromatinization can influence the global methylation level. However, such dependency did not apply to each studied Secale taxon. Gathering of various types of sequences may result in the increase of the DNA quantity in the genome but, as not all of them are highly methylated, the differences in the DNA methylation level are not surprising. Additionally, various events during the evolution of Secale taxa could affect their epigenomes. The results obtained in this study are evidence that the percentage of methylated cytosine cannot be inferred solely based on the genome size or t-heterochromatin. This is a significantly more complex issue.

Supplemental Information

Supplemental Information 1 ELISA methylation data

Click here for additional data file.

Supplemental Information 2 Flow cytometry data

Click here for additional data file.

Supplemental Information 3 MSAP data

Click here for additional data file.

Supplemental Information 4 t-heterochromatin Secale data

Click here for additional data file.

The authors are very grateful to prof. Jolanta Tarasiuk and Dr Agnieszka Maruszewska for their help in the flow cytometric analysis. They also wish to thank Dr Anetta Wieczorek for her help in the statistical analysis.

Abbreviations

MSAP Methylation-Sensitive Amplification Polymorphism

ELISA enzyme-linked immunosorbent assay

MSL methylation-susceptible loci

NML non-methylated loci

PCoA Principal Coordinate Analyses

IRAP inter-retrotransposon amplified polymorphism

TE transposable elements

MOPS 3-(N-morpholino)propanesulfonic acid

TBE Tris-borate-EDTA

APS ammonium persulfate

TEMED 1,2-Bis(dimethylamino)ethane.

Additional Information and Declarations

Competing Interests

Author Contributions

Data Availability

The authors declare there are no competing interests.

Anna Kalinka conceived and designed the experiments, performed the experiments, analyzed the data, contributed reagents/materials/analysis tools, wrote the paper, prepared figures and/or tables, reviewed drafts of the paper.

Magdalena Achrem performed the experiments, analyzed the data, contributed reagents/materials/analysis tools, wrote the paper, reviewed drafts of the paper.

Paulina Poter performed the experiments.

The following information was supplied regarding data availability:

The raw data has been supplied as a Supplementary File.

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
