# Peer review of "The DNA methylation level against the background of the genome size and t-heterochromatin content in some species of the genus Secale L"

_PeerJ, doi:10.7717/peerj.2889_

## Round 0.1 · original submission · Minor Revisions

In general, reviewers considered the manuscript to be well-written; however, I would consider some of the comments provided by the third reviewer to appease some of the disagreements in presentation. A manuscript that will reach many types of readers work more toward building a well structured and well received paper. There is no real page limitation; however, you may want to consider the relevance of some of the indicated text by one reviewer; I feel that it is fine.

Perhaps some additions to the discussion might include whether some of the methods here be compared to that of barley or wheat diploid progenitor species considering that the chromosome structures may be related and that sequence data is close to being available. Can any recent rye sequence maps also lead to comparisons with this work?

It would be valuable to consider the recommendation suggested by one reviewer to compare and contrast findings with regard to the referred to and referenced publication.

Additional input which might be helpful would be to provide an example of a silver-stained gel from the MSAP method? Or perhaps a sample Giemsa stained preparation?

It appears extended efforts have been carried out to help add information to a forming hypothesis for Secale spp. genome structure; however, it seems more questions have arisen. For future work that may extend this work, this is perhaps a thorough starting point. I will suggest that this manuscript requires some minor revisions. Please try to address the provided suggestions. Thank you for providing this manuscript for review.

Additional suggested edits:

LINE NO.: / PREVIOUS FORM / SUGGESTED FORM / [ADDITIONAL NOTES]
92: / S. sylvestre has the lowest / Secale sylvestre has the lowest / [ full Genus name extended at start of a sentence. ]
94: / S. cereale is considered / Secale cereale is considered / [ full Genus name extended at start of a sentence. ]
257: / S. vavilovii also showed / Secale vavilovii also showed / [ full Genus name extended at start of a sentence. ]
316: / Secale c. ssp. segetale / S. c. ssp. segetale / [ Genus may be possibly abbreviated here. ]
441: / S. s. ssp. africanum / Secale strictum ssp. africanum / [ full Genus/species name extended at start of a sentence. ]

Reviewer 1 ·

Basic reporting

No Comments.

Experimental design

No Comments.

Validity of the findings

No Comments.

Additional comments

In this manuscript, authors tried to elucidate the relationship between genome size and DNA methylation level for some species of the genus Secale L. by using ELISA and MSAP to detect global and specific sites of genomic methylation, and also used Giemsa staining and flow cytometry to detect t-heterochromatin contents and genome size. Their major conclusion was that the percentage of methylated cytosine cannot be inferred solely based on the genome size or t-heterochromatin and the relationship between genome size and DNA methylation level was more complicated. Authors acknowledged limitations of the methods used and they gave the conclusions a reasonably conservative discussion. Nevertheless, relative to the Result section the discussion is still too long with some contents not directly relevant to the results. This manuscript in general was well written, the data obtained were reliable and analysis was appropriate. I suggest (1) to further shorten the Discuss section; (2) to conduct a more careful proof-reading.

Reviewer 2 ·

Basic reporting

No Comments

Experimental design

No Comments

Validity of the findings

No Comments

Additional comments

This paper is not written in the appropriate way. There are a lot of formatting errors.
I don’t understand why the authors performed all the statistical analyses, and what a null distribution is. In almost all cases, there is no significance, and what? There is not enough citations. The aim of the research and conclusion are ambiguous.

Introduction
In line 83-86, as for the phylogenetic information of genus Secale, the ms said, “Despite years of research on the genus Secale, they have not been fully determined”. However, in line 112-115, it said, “The aim of the research was to answer the question whether the cytosine methylation variation was consistent with phylogenetic relationships”. If the phylogeny of the genus Secale is not resolved based on genetic variation, how do the authors test the consistency of the phylogenies based on methylation variation? No one knows the correct phylogeny.

In line 87-89, “Hence, there is a need to search for different methods to verify the proposed systems of classification and phylogenetic relationships”.
I don’t think methylation differences are appropriate for revealing the phylogenetic relationship. First, when focusing on a single cytosine site, DNA methylation is highly variable even among very closely related individuals. This case is very similar to the situation, in which tons of back mutations have occurred that reduce the resolution of phylogenetic relationships. Second, it is well known that AFLP-based approaches are not appropriate for inferring phylogenies because of a large number of homoplasy. So, it is NOT good to set the research purpose as to reveal the phylogenetic relationship of the genus Secale.

I’m not a specialist of Secale sp., and felt that there is a few information about these species, e.g., divergence time or nucleotide divergence, chromosome numbers, classification etc. It is better to add an additional figure for the summary of the previous phylogenetic studies based on nucleotide sequences.

Some studies have revealed that the correlation between methylation levels and genome size, but there is no citation in Introduction.

A lot of citations are missed as for epigenetics and methylomes.

Methods
Line 182-189: First, the equation is unreadable. Second, Wang et al. [93] should be [92] (I found a lot of this kinds of mistakes that frustrated me!!). Third, I cannot understand what “differences between the taxons” is (also, “taxons” should be “taxa”). I guess the difference of methylation levels between species, but I’m not sure.

Results
Line 258-259: Fig. 4 is called, but it does not indicate “total methylation level of the studied loci was very similar in all taxa”.

Line 265: “methylation pattern did not differ significantly between any pair of taxa (Table 3).” Again, I don’t understand what the U test means. Does the lower P-value means highly divergent methylation patterns? In Table 3, 0,95 should be 0.95 and so on.

Line 271-272: I’m so annoyed. Where are Fig. 2a, 2b and 2c?

Line 276: Fig. 3 shows the result of PCA, not dendrogram. I don’t like this ms. I can’t understand which figure is called!!

Line 287-288: “No significant differences in the content of t-heterochromatin were found between these taxa (Table 4)”. As in MSAP analysis, why did the authors think “significant difference” is so important? Again, the authors observed no significance. If so, this result should not be published.

Line 291-292: “Therefore, these Secale taxa could be divided into two groups, with a relatively lower and higher content of t-heterochromatin”. Meaningless treatment.

Line 321: “CNG and CNN sequence methylation also occurs”. Add more appropriate citations.

Line 322-: “Thus, it was possible that the CpG methylation was similar in the genomes of all Secale taxa tested, while such high differences in the level of 5-methylcytosine were due to differences in the CNG and CNN methylation”. This statement is speculative. Should be verified by bisulfite sequence.

Figure 7 or 5: The last figure (also other figures) are labeled by 7 and 5. What is shown on the x-axes?

Reviewer 3 ·

Basic reporting

This research article reports on experimental approaches to identify interspecific variation of cytosine methylation and its relationship with the size of the genome (C-value) and the content of telomeric heterochromatin in different species of the Genus Secale.
The report is clear, unambiguous, and used professional English language throughout the manuscript.The introduction provides comprehensive informations on the background to show the context of the research. The literature is generally well referenced and relevant. However, a previously published report by Fulneček and Kovařík (2014) on the challenges of the interpretations on MSAP profiles has not been considered by the authors.
The structure of the manuscript conforms to PeerJ standard discipline norm. The figures are relevant, well labeled and described, however, quality of fig. 3 needs to be improved with respect to the resolution. It appears to be, that the figure is reproduced only in part and information on S. strictum (green ellipse) is missing?

Experimental design

The manuscripts reports on original primary research within the scope of the journal. The research question is well defined, relevant & meaningful. It is stated how research fills an identified knowledge gap. The investigation have been performed to a high technical & ethical standard. The Methods described with sufficient detail & information to replicate.

Validity of the findings

The impact and novelty is assessed and the negative result, e.g. that the methylation pattern did not differ significantly between any pair of taxa, is accepted. It needs to be specified, whether for the MSAP analysis three technical or biological replicates have been used (l. 167). As the MSAP analysis may introduce some errors (l. 234), only biological replication can be accepted as meaningful.
The data are robust, statistically sound, and controlled within the applied experiemental design. However, the authors did not implement recent progress to in improve interpretation of MSAP patterns (cv. Fulneček and Kovařík 2014). The conclusion drawn by the authors are well stated, linked to original research question & limited to supporting results.

Additional comments

Reference

Fulneček J, Kovařík A (2014) How to interpret methylation sensitive amplified polymorphism (MSAP) profiles? BMC Genet. 15:2. doi: 10.1186/1471-2156-15-2.

---

## Round 0.2 · accepted · Accept

Thank you for considering the suggested edits to the manuscript. Your addition of the figure was helpful. Considering that this may be a platform to expand methylation analysis and perhaps address different hypotheses on variable genome content I feel that your initial survey serves the research well for expanding such studies. As key cereal genome sequences become available perhaps additional questions will be raised. A few minor edits; however, not critical would be to remove the dashes ('-') at each of the sections of the MATERIALS AND METHODS. Thus, I will approve this as is and let the production team to decide on the proper formatting. Thank you for the manuscript submission and I am clearing this for movement toward publication. Congratulations.